# Mathematical modeling and parameter estimation of levodopa motor response in patients with parkinson disease

**Mauro Ursino**[1]*, **Elisa Magosso**[1°], **Giovanna Lopane**[2,3°], **Giovanna Calandra-Buonaura**[2,3‡], **Pietro Cortelli**[2,3‡], **Manuela Contin**[2,3°]

**1** Department of Electrical, Electronic and Information Engineering, University of Bologna, Cesena, Italy, **2** IRCCS, Istituto delle Scienze Neurologiche di Bologna, Bologna, Italy, **3** Department of Biomedical and Neuromotor Sciences, University of Bologna, Bologna, Italy

☯ These authors contributed equally to this work.
‡ These authors also contributed equally to this work.
* mauro.ursino@unibo.it

**Data Availability Statement:** All relevant materials underlying this study has been made available. The clinical protocol used for Levodopa kinetic-dynamic test is available in protocols.io (Levodopa

## Abstract

Parkinson disease (PD) is characterized by a clear beneficial motor response to levodopa (LD) treatment. However, with disease progression and longer LD exposure, drug-related motor fluctuations usually occur. Recognition of the individual relationship between LD concentration and its effect may be difficult, due to the complexity and variability of the mechanisms involved. This work proposes an innovative procedure for the automatic estimation of LD pharmacokinetics and pharmacodynamics parameters, by a biologically-inspired mathematical model. An original issue, compared with previous similar studies, is that the model comprises not only a compartmental description of LD pharmacokinetics in plasma and its effect on the striatal neurons, but also a neurocomputational model of basal ganglia action selection. Parameter estimation was achieved on 26 patients (13 with stable and 13 with fluctuating LD response) to mimic plasma LD concentration and alternate finger tapping frequency along four hours after LD administration, automatically minimizing a cost function of the difference between simulated and clinical data points. Results show that individual data can be satisfactorily simulated in all patients and that significant differences exist in the estimated parameters between the two groups. Specifically, the drug removal rate from the effect compartment, and the Hill coefficient of the concentration-effect relationship were significantly higher in the fluctuating than in the stable group. The model, with individualized parameters, may be used to reach a deeper comprehension of the PD mechanisms, mimic the effect of medication, and, based on the predicted neural responses, plan the correct management and design innovative therapeutic procedures.

## 1. Introduction

Parkinson disease (PD) is a chronic, progressive neurodegenerative disease characterized by bradykinesia in combination with either rest tremor, rigidity or both, that typically shows a

kinetic-dynamic test, M. Contin) at http://dx.doi.
org/10.17504/protocols.io.bbq8imzw. The
neurocomputational model of basal ganglia,
levodopa pharmacokinetics and
pharmacodynamics, as well as the parameter
estimation procedure are written in Matlab. Matlab
codes and data are available in ModelDB (Basal
Ganglia and Levodopa Pharmacodynamics model
for parameter estimation in PD (Ursino et al 2020))
at http://modeldb.yale.edu/261624.

**Funding:** The author(s) received no specific
funding for this work.

**Competing interests:** The authors have declared
that no competing interests exist.

clear and dramatic beneficial response to dopaminergic treatment [1]. Motor signs can be mainly ascribed to an impaired response selection, preparation or inhibition in the basal ganglia (BG), due to a malfunctioning of the Go/NoGo balance in the direct and indirect pathways of the striatum. Malfunctioning of BG, in turn, is secondary to an insufficient dopamine production by the Substantia Nigra pars compacta. Dopamine, in fact, has an excitatory role in the Go path, thus favouring action selection, whereas its depletion favours the NoGo activity.

Accordingly, dopaminergic treatment, basically administration of the dopamine precursor levodopa (LD), reliably ameliorates the classic motor symptoms in PD patients. However, with disease progression and longer LD exposure, patients develop LD–induced motor complications, like wearing off that is a re-emergence of parkinsonism related symptoms at the end of LD-dose or unrelated to LD-dose. The practical benefits of LD in advanced disease stage are hindered by modifications in drug kinetic and dynamic mechanisms, resulting in a fluctuating response during the day.

In association with motor fluctuations the use of higher doses of dopamine are often associated with dyskinesia, involuntary movements that can occur at peak effect of LD dose, at the beginning and end of dose, or between doses [2,3].

A typical test to assess the motor symptoms of PD (especially bradykinesia) is the alternate finger tapping test. Several evidences have shown a significant correlation between results of this task and part III of the Unified Parkinson's Disease rating scale, namely the bradykinesia sub score [4,5]. Moreover, alternate finger tapping test seems appropriate for the evaluation of LD effects on bradykinesia, and to discriminate between patients with stable or fluctuating response to the pharmacological treatment [6].

Most previous quantitative studies on the finger tapping response, and on its relationship with LD intake, however, have been performed using empirical equations, which do not enter into the complex neural mechanisms at the basis of action selection in the basal ganglia [3,7–9]. On the other hand, various authors [10–13] underlined the necessity to incorporate neural mechanisms in any quantitative assessment of PD, not only to account for the dependence of striatal neurons response on D1 and D2 dopamine receptors (i.e., the so called dopamine "gain factor"), but also to consider the complex corticostriatal plasticity (i.e., "learning factors"). An influential hypothesis is that the behaviour of the patient depends, beside on the actual dopamine level (and so, on the present response to the LD treatment) also on synaptic plasticity which, in turn, is associated to motivational processes, reward and reinforcement. According to this idea, abnormal plasticity and aberrant learning may be significant but still unrecognized components of current drug therapies [11,13].

The latter aspects can be clarified with the use of neuro-computational models, which incorporate both the LD pharmacokinetics and pharmacodynamics, and the neural mechanisms involved in the selection of motor response. The use of these models may endorse a deeper quantitative understanding of bradykinesia and dyskinesia, at different PD stages, and clearer formalization of therapeutic targets.

Unfortunately, most models presented until now, although of great scientific value [14–19], do not incorporate a clear relationship between pharmacodynamics and neural responses, nor are built to analyse the finger tapping test in a direct, straightforward way, including neural factors in the BG.

Recently, we built a simple but complete model of action selection in the BG, which incorporates the most important neural and pharmacodynamical mechanisms involved. The neuro-computational model simulates the activity in the direct, indirect and hyperdirect pathways of the BG, their dependence on dopamine levels (with a distinction between D1 and D2 receptors), the role of cholinergic mechanisms, and the Hebbian mechanisms of corticostriatal synapse potentiation and depotentiation [20]. In particular, each neuron in the striatum is

modelled with a first order dynamics and a sigmoidal relationship (to simulate the overall activity range, from silence to upper saturation), where the input to the sigmoid depends both on the activity of pre-synaptic cortical neurons and on a "dopaminergic input" (the latter is excitatory in the Go pathway and inhibitory in the NoGo one). A recent version of this model [21], considering just the selection of two simple movements, was used to simulate the alternate finger tapping response after LD administration. To this end, LD pharmacokinetics was simulated assuming a three-compartmental model (the first compartment represents a central pool, where LD is administered and its concentration measured in plasma, the second is a peripheral pool, describing the interaction between plasma and other body fluids, and the third represents dopamine concentration in the basal ganglia). In order to mimic how LD can affect the striatal neurons [21], the dopaminergic input of the striatal neurons monotonically depends on dopamine concentration in the BG compartment via the Hill law. This means that higher values of dopamine concentration in the BG have a stronger effect on striatal dopaminergic input, until a saturation is reached.

In the previous paper [21], parameters of the pharmacodynamical model were manually adjusted, to simulate the finger tapping frequency of six patients (three stable and three with a fluctuating response) in the 4 hours after LD administration, just to show model's capacity to simulate a variety of individual cases.

Aim of the present paper is to significantly improve the previous results: i) by implementing an automatic procedure for parameters assignment in the individual patient; ii) to test the presence of statistically significant differences in parameters of the two groups to unmask the pharmacodynamics mechanisms involved; iii) to illustrate, with a few exempla, how the neuro-computational model, with patient-specific pharmacodynamical parameters assigned, can be used to analyse some aspects of patient response, such as synaptic plasticity, at single subject level.

## 2 Method

### 2.1 Qualitative model description

An accurate description of the model, with all equations and parameter numerical values is provided in previous works, where all details can be found [22]. Hence, only the main qualitative elements are given below. A complete list of equations can also be found in the S1 Material.

A summary of the different model parts is presented in Fig 1a.

**Model of LD pharmacokinetics and pharmacodynamics.** The kinetics of LD was simulated with a three-compartment model: as specified above, the first represents a central pool, the second is a peripheral compartment, and the third is an "effect compartment". This part of the model contains seven parameters, the meaning of which is reported in Table 1.

The LD pharmacodynamics mimics how the concentration in the effect compartment (named $c_3$ in the following) modulates the activity of the Go and NoGo neurons in the neuro-computational model.

First, in order to account for the observed delay between LD concentration in plasma and its motor effect, we introduced a pure delay, $T$. By denoting the delayed concentration as $c_{3delay}$, we can write:

$$c_{3delay}(t) = c_3(t - T) \tag{1}$$

where $t$ represents the present time.

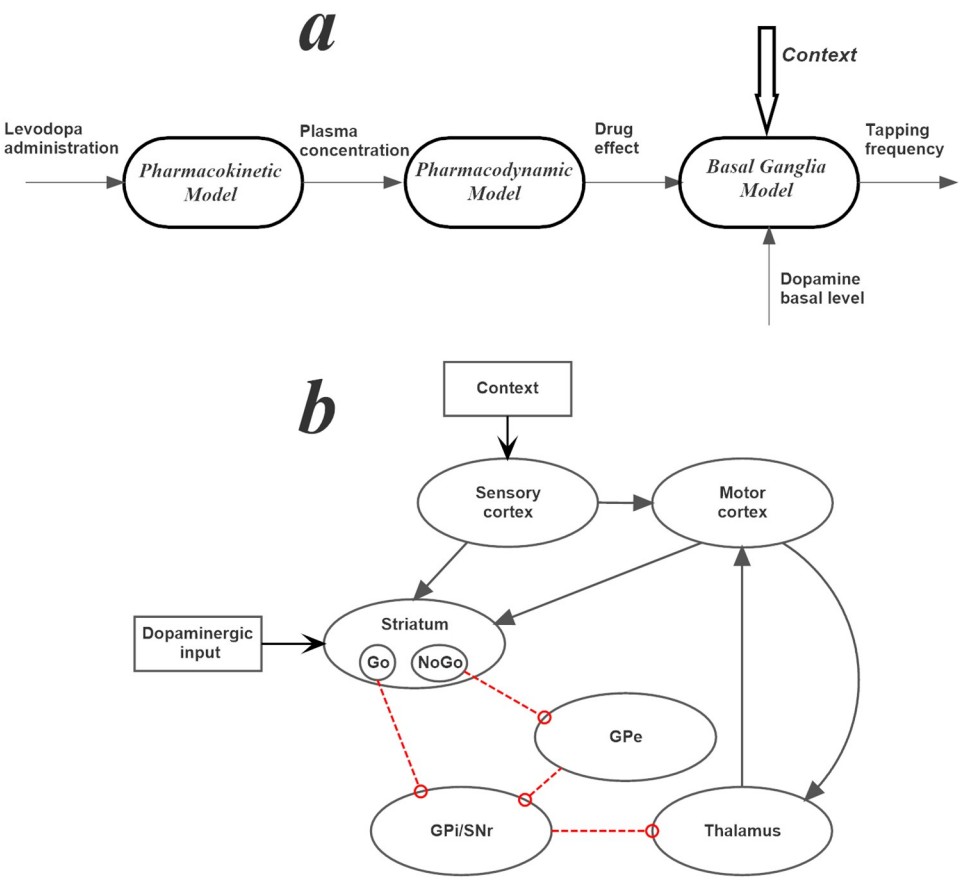

**Fig 1.** a) General structure of the mathematical model used in this work. The first part (pharmacokinetics) is described via a two-compartmental model of LD concentration in plasma. The second (pharmacodynamics) describes the concentration of the drug in the brain and its effect on striatal neurons. The third (neurocomputational model) describes action selection by the basal ganglia during a finger tapping test, as a function of the dopaminergic input. b) Topological organization of the neurocomputational model, where black lines with arrows denote excitatory connections, and red lines with open circles inhibitory connections. GPe: Globus Pallidus pars externa; GPi/SNr: Globus Pallidus pars interna/Substantia Nigra pars reticulata.

**Table 1. Meaning of the main pharmaco-dynamical parameters.**

| *Pharmaco-kinetics* | |
|---|---|
| $k_{12}$ and $k_{21}$ | inter-compartment rate constants; |
| $k_{12}$ and $k_{21}$ | inter-compartment rate constants; |
| $k_{etot} = k_{e1} + k_{31}$ | total body rate constant; |
| $V_1$, $V_2$ and $V_3$ | compartment volumes; |
| *Pharmaco-dynamics* | |
| $k_{e3}$ | drug removal from the effect compartment; |
| $D_0$ | basal value of the dopaminergic term, immediately before LD administration |
| $T$ | time delay for the LD effect in the compartment |
| $D_{max}$ | maximum effect that LD can produce |
| $D_{c50}$ | concentration in the effect compartment which produces 50% of the maximum effect |
| $N_D$ | Hill coefficient, which determines the slope of the concentration-effect relationship |

Binding of dopamine with a receptor has been described with the Hill law [23]:

$$D(t) = D_0 + \frac{D_{max} c_{3delay}^{N_D}(t)}{D_{c50}^{N_D} + c_{3delay}^{N_D}(t)} \tag{2}$$

where the quantity $D$, named "dopaminergic input" in the following, represents the effect that a LD concentration $c_3$, delayed by a time T (see Eq 1) induces on the Go and NoGo neurons in the BG neurocomputational model, and the meaning of the other parameters is reported in the second part of Table 1.

In order to mimic the temporal pattern of LD concentration in plasma in the individual patient, three parameters of the pharmacokinetic model (i.e. $k_{12}$, $k_{21}$, $k_{etot}$) have been estimated; moreover, six parameters of the pharmacodynamics ($k_{e3}$ and the parameters in the Hill law, $D_0$, $T$, $D_{max}$, $D_{c50}$, and $N_D$) have been estimated to mimic the tapping frequency after LD administration, i.e., the motor effect of the drug. The estimation procedure is described in section 2.3.

**The neurocomputational model.** This part of the model receives the "dopaminergic term" ($D(t)$ in Eq 2) as input, and mimics the alternative motor selection during the tapping test. Its actual output is the tapping frequency (in taps/min).

Computational units in the model are represented via firing rate neurons, which receive the sum of excitatory or inhibitory inputs coming from pre-synaptic neurons, weighted by their mutual specific synaptic strengths, and produce their output via a low-pass first order filter and a sigmoidal relationship. The topological structure of the network (summarized in Fig 1b) mimics the present knowledge on the basal ganglia (BG) [16,24–28]. Briefly, a sensory representation of the external stimuli in the cortex is transmitted to the motor cortex where a winner-takes-all dynamics is implemented to select the appropriate action. Each action is represented via a segregated channel in competition with the others. Competition is realized via lateral inhibitory synapses in the cortex, and a positive feedback from the thalamus. The information on possible action selection is then transmitted from the cortex to the cells in striatum (each action via a segregated channel). These cells are divided into two alternative classes depending on their dopamine response and their efferent projections. The Go cells project to the Globus Pallidus pars interna and Substantia Nigra pars reticulata (GPi/SNr); excitation of the Go cells has the effect of disinhibiting the thalamus, thus favoring action selection. The NoGo cells project to the Globus Pallidus pars externa (GPe), and then indirectly to the GPi/SNr; excitation of the NoGo cells has the opposite effect of inhibiting the thalamus and blocking the action selection. The last part of the model is the thalamus, which closes the loop with the motor area of the cortex. Hence, neurons in the thalamus, coding for different actions, are normally inhibited by the GPi/SNr. Only when a Go pathway prevails on the NoGo within an action channel, the corresponding neuron in the thalamus is dishinibited, thus exciting the neuron in the motor cortex.

The activity in the Go and NoGo pathways is affected by the level of dopamine (or LD) expressed through the quantity D in Eq (2). D is primarily inhibitory for all the NoGo neurons, and excitatory for the winning Go neuron. Furthermore, the effect of the dopaminergic quantity D is potentiated by the role of cholinergic neurons [21,29–31].

The alternate finger tapping task was simulated as a choice between two actions (finger down in either position, finger up and shift) which are alternatively given as input to the model. The tapping frequency depends both on the status of corticostriatal synapses (which is set at a fixed value in this work throughout the parameter fitting procedure; a sensitivity analysis is performed in the last portion of section Results) and on the level of the dopaminergic

input (D(t) in Eq 2, depending on LD kinetics and dynamics whose parameters are estimated here on the basis of clinical data, see Section 2.3).

BG can also be trained, considering the presence of rewards and punishments. To this end, we assumed that the synapses entering into the Go and NoGo striatal neurons can be potentiated or depotentiated with a Hebbian mechanism; in particular, phasic bursts of dopamine (rewards) enhance cortico-striatal synaptic plasticity in the Go pathway, whereas drops in dopamine (punishments) strengthen cortico-striatal synapses in the NoGo [32–37].

## 2.2 Description of the clinical procedure

**Patients.** LD kinetic-dynamic results were obtained from 26 PD patients referred to the Institute of Neurological Sciences of Bologna for therapeutic drug monitoring (TDM). PD patients were recruited within the "Bologna Motor and Nonmotor Prospective Study on Parkinsonism at Onset" (BoProPark study, protocol number 09070, approved by the Independent Ethics Committee of Bologna Health Authority. Kinetic-dynamic modelling of LD motor response was part of the study. Patients gave their written informed consent to participate in the study and to publish the data.

Patients were grouped according to their subjective clinical response to LD, neurologist's repeated observations and objective LD test dose motor assessment as: "stable" patients, with no clinically appreciable diurnal changes in motor performance, receiving a minimum of two up to three doses per day; "fluctuating" patients, showing predictable fluctuations in motor state ("wearing off" phenomena), clearly related to levodopa dosing, three to six times a day. Clinical characteristics of patients are reported in Table 2. The two patients' groups were comparable for age, weight, sex and significantly different (p<0.001) for Hoehn and Yahr and UPDRS score, PD symptoms' and LD therapy duration, LD daily dose and LD equivalent daily dose (LEDD) [38].

**LD kinetic-dynamic test.** On the morning of TDM the patients received an oral fasting dose of LD/benserazide (100/25 mg) after a 12 hours' washout of LD and concomitant antiparkinsonian drugs.

**Table 2. Patients' characteristics at levodopa test.**

|  | Stable (n = 13) | Fluctuating (n = 13) |
|---|---|---|
| Age (year) | 63±9 | 65±9 |
| Sex (m/f) | 8/5 | 7/6 |
| Weight (kg) | 78±18 | 73±13 |
| Parkinsonian symptoms' duration (year) | 1.8±0.7 | 6.1±2.9 |
| Hoehn and Yahr score | 1.5 (1–2) | 2 (2–2.5) |
| UPDRS score | 15±5 | 29±8 |
| LD therapy duration (year) | 0.6±0.4 | 4.4 ± 2.7 |
| LD daily dose (mg/d) | 208±27 | 381±103 |
| Concomitant antiparkinsonian drugs | Pramipexole (n = 3) | Amantadine (n = 1) |
|  | Rasagiline (n = 1) | Pramipexole (n = 3) |
|  |  | Ropinirole (n = 4) |
|  |  | Rasagiline (n = 2) |
| LEDD (mg/d) | 240±67 | 455±125 |

Data are expressed as mean±standard deviation or median (25th-75th percentiles).

m = male; f = female; kg = kilogram; UPDRS, Unified Parkinson Disease Rating Scale; LD = levodopa; mg = milligram; d = day; LEDD, levodopa equivalent daily dose, according to Tomlinson et al. [38].

Blood venous samples for LD plasma concentration analysis were drawn by an indwelling catheter immediately before the LD dose, at 15-minute intervals for the first 90 minutes, then half-hourly up to 3 hours after dosing [39].

Patients' motor responses to the LD test dose were simultaneously assessed by the alternate index finger tapping test (the number of times the patient could alternately tap two buttons 20 cm apart in 60 seconds with the most affected hand) up to 4 hours post-dosing, using a computerized touch screen system (mTapping, mHealth Technologies srl, Bologna, Italy) [39].

Latency to onset of a clinical significant motor response elicited by the LD test dose is defined as the time to increase in tapping frequency of ≥15% of baseline values. Duration of the tapping effect was calculated as the difference between the time to return to < 15% of baseline values and time to onset of response.

Patients of the fluctuating group and patients of the stable group (see above) presented return and no return to baseline tapping performance, respectively, within the 4 hours' length of examination.

## 2.3 Parameter estimation procedure

All parameters in the basal-ganglia neurocomputational model have been assigned a priori, to simulate a typical network for a PD subject (see [22]), assuming that PD patients have less-developed cortico-striatal synapses in the Go and NoGo pathways compared with an healthy subject, as a consequence of reduced training capacity.

Conversely, parameters in the pharmacokinetic and pharmacodynamics equations have been estimated on the individual patient, to minimize differences between measured and simulated data after LD administration. Hence, differences among patients in the present study are only ascribed to changes in LD pharmacodynamics, not to neural network characteristics.

We adopted this choice for two reasons

1. We wish to assess which parameters in the LD pharmacodynamics have a major role to discriminate between stable and fluctuating patients;

2. Inclusion of additional aspects in the fitting procedure would make the number of estimated parameters too large compared with the available data, inducing a high risk of overfitting (see [40]). We are aware that some parameters in the neural network deserve individual estimation too (see also [22] for a sensitivity analysis on the neurocomputational model), but this requires the use of further data on neuro-motor aspects of the patients to avoid overfitting. Such a problem may be considered in future evolutions of this study. However, a preliminary analysis on the possible effect of synapse changes on the estimated parameters is reported in the last part of section Results.

Parameters in the pharmacokinetic models ($k_{12}$, $k_{21}$, $k_{etot}$) were estimated for each patient, to mimic the temporal pattern of LD concentration in plasma. It is worth noting that parameters $V_1$, $V_2$, and $V_3$, representing the compartment volumes, were not estimated individually, but we used prototypical values. In fact, estimation of three parameters was always sufficient to reach a good reproduction of LD concentration in plasma, in all examined patients. Estimation was achieved by minimizing the sum of square errors between model and measured plasma concentration during three hours after administration. Minimization of the cost function was achieved with the Nelder-Mead algorithm [41] which does not require the computation of the gradient. We used a single "initial guess" for the parameters in all patients: ($k_{12}$ = 1.5 l/min; $k_{21}$ = 1.5 l/min; $k_{etot}$ = 3 l/min).

Estimation of the pharmacodynamics parameters ($k_{e3}$, $D_0$, $T$, $D_{max}$, $D_{c50}$ and $N_D$) was much more complex and required a more sophisticated strategy. First, the value of LD concentration

in plasma, obtained in each patient from the previous optimization procedure, was given as input to the pharmacodynamics part of the model (consisting in the effect compartment + the Hill law), to compute the temporal pattern of the "dopaminergic term" ($D(t)$ in Eq 2) to be passed to the neurocomputational BG model. It is worth noting that $D(t)$ depends on the pharmacodynamics parameters to be estimated. Finally, the neurocomputational model uses $D(t)$ as an input, to evaluate the temporal pattern of the finger tapping frequency.

As specified above, we did not individually estimate the parameters in the neural network, but we just estimated the pharmacodynamics parameters. To this end, starting from an initial guess, we minimized the following cost function of the difference between the tapping frequency evaluated as output of the neural net and the values measured in the patient

$$F(\theta) = \sum_{i=1}^{M} \left[ f_{\text{mod}}(t_i, \theta) - f_{\text{meas}}(t_i) \right]^2 + k \max_i \{ |f_{\text{mod}}(t_i, \theta) - f_{\text{meas}}(t_i)| \} \tag{3}$$

where $t_i$ represent the i-th measurement instant, $M$ is the number of measurements, $f_{\text{mod}}(t_i, \theta)$ is the tapping frequency predicted by the model at the instant $t_i$, which is a function of the parameter $\theta$, and $f_{\text{meas}}(t_i)$ is the tapping frequency measured on the patient at the instant $t_i$. The first term in the right-hand member of Eq (3) is the classic sum of square errors, while the second term is the maximum absolute error, weighted by a factor $k$. The latter term was included to better simulate the rapid decrease in tapping frequency observed in unstable patients during the end of the observation period. Eq (3) was minimized with the Nelder-Mead minimization algorithm. A good fitting was achieved using $k = 10$. A schema of the fitting procedure can be found in Fig 2.

Moreover, since results of the minimization procedure depends on the initial guess for the parameters (i.e. the cost function in Eq (3) exhibits many local minima) the minimization procedure was repeated with 10 different values of the initial guess, randomly chosen, and the best result (i.e., that associated with the deeper minimum of the cost function) was chosen. Only the initial guess value of parameter $D_0$, which establishes the basal value for the tapping frequency before LD infusion, was not randomly assigned, but computed to have the same initial tapping frequency as the patient. This parameter, however, was then estimated as the other five parameters by the Nelder-Mead algorithm, assuming that also the basal tapping frequency may be affected by error.

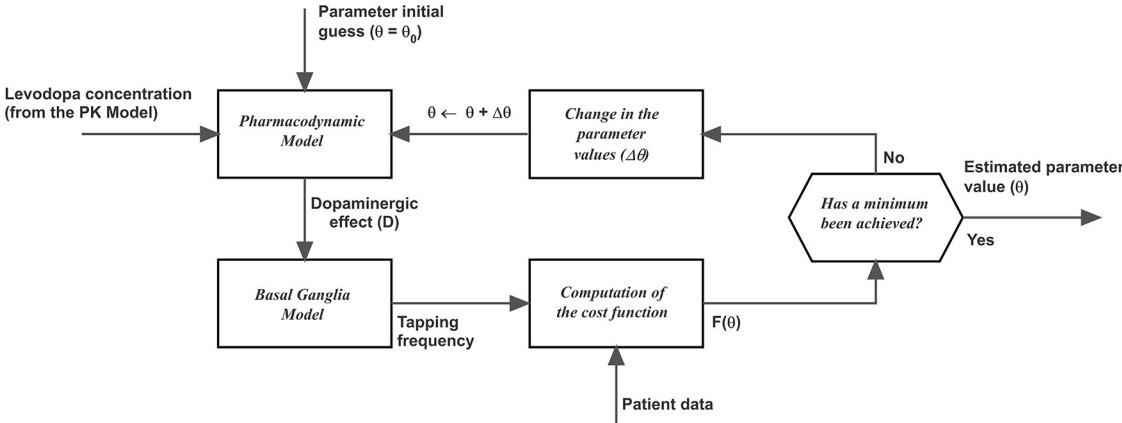

**Fig 2. Algorithm used in the model to automatically estimate the parameters of LD pharmacodynamics, via minimization of a cost function.** It is worth-noting that a similar algorithm has been previously used to estimate the pharmacokinetics (PK) parameters too, to model the temporal pattern of plasma dopamine concentration.

The overall goodness of fitting was evaluated by computing the $R^2$, separately in the two classes. $R^2$ measures the amount of variance in the data that can be explained by the model. Values higher than 0.8 are generally considered as a good fitting.

The estimated parameters for pharmacokinetic and pharmacodynamics were compared in the two groups of patients (stable and fluctuating) to test for statistically significant differences. Since some variables did not pass the Gaussianity test (performed with the omnibus K2 test) pairwise comparisons among parameters were performed via Wilcoxon signed-rank tests with Bonferroni correction.

Finally, the temporal pattern of tapping frequency was further simulated in two patients (one of the first group and one of the second), using the values of pharmacodynamics parameters previously estimated, but assuming different synapse levels in the striatum, to assess the robustness of the results vs. changes in the neurocomputational model.

## 3. Results

The relationship between the dopaminergic input (quantity $D$ in Eq 2) and the taps frequency, as obtained with the BG neural network model, is illustrated in Fig 3. This figure is a little different from that shown in a previous paper (see Fig 3 in [22]) since we used smaller values for the neuron time constant ($\tau$ = 10 ms instead of 15 ms) and we used a smaller time lag between the two finger movements (90 ms instead of 115 ms). These changes have been adopted since,

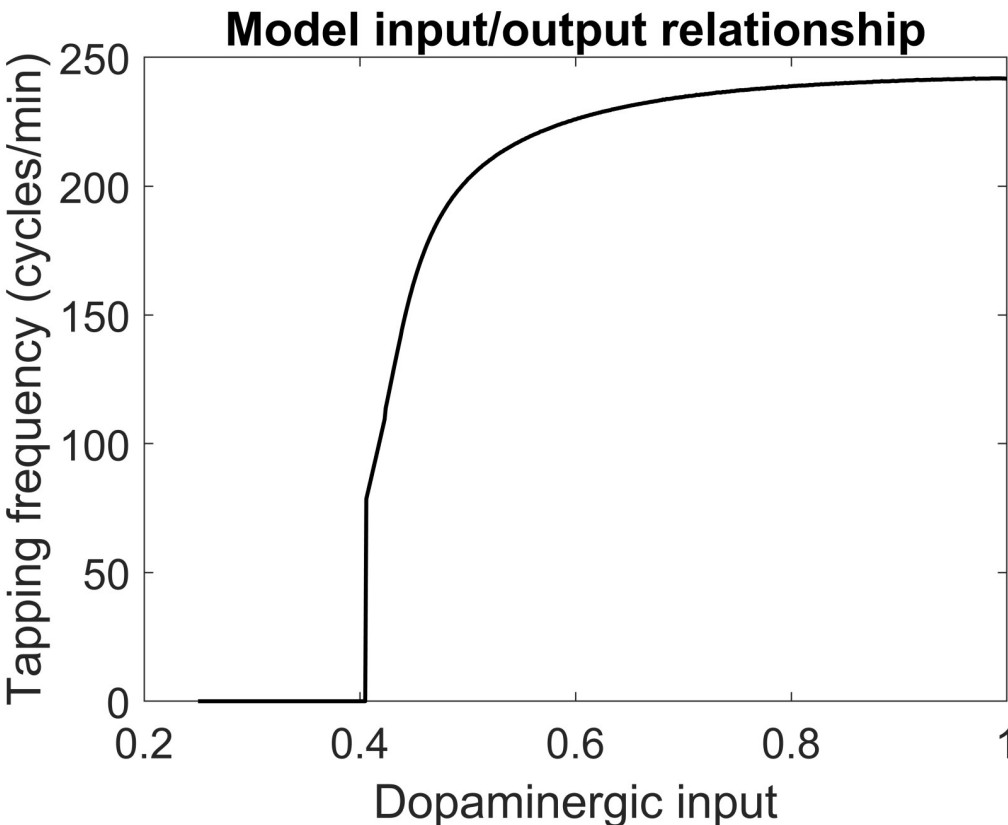

**Fig 3. Steady-state relationship between the dopaminergic input (D) and tapping frequency according to the basal ganglia neurocomputational model.** It is worth-noting that this relationship reaches higher values compared with that used in a previous paper [22] to provide better estimation of patients' data. This has been achieved by reducing the neuron time constant from 15 to 10 ms, and the time lag between the two movements from 115 to 90 ms.

with the previous values, the maximum tapping frequency was approximately 180 cycles/min; conversely, a few patients in the present study reach tapping frequency values, after LD oral dosing, as high as 220–230 cycles/min, as in Fig 3.

## Results of the fitting procedure

As explained in the Method session, patients had been previously subdivided into two groups (stable and fluctuating). Patients in the first group can maintain an increased finger tapping frequency (hence, an improvement in bradykinesia) for several hours after the administration; those in the second group exhibit a post-improvement return to baseline performance within the three-four hours of LD dosing. Data in both groups have been fitted with the same model and with exactly the same fitting procedure, as described above. Hence, differences can only be ascribed to the final estimated parameter values.

Two examples of fitting, in patients of the first group (stable) are illustrated in Fig 4, while two examples of fitting in patients of the second group (fluctuating) are shown in Fig 5. In both figures the left panels show LD concentrations in plasma while the right panels show the tapping frequency. The dashed red (or magenta) lines with asterisks (or crosses) represent patient data, while the continuous blue (or green) lines with open circles (or triangles) are model simulation results obtained with optimal parameter values (i.e., with the parameters obtained through the best fitting procedure).

The figures show that the pattern of LD concentration in plasma is quite similar in the different patients. Conversely, greater differences are evident in the pattern of tapping frequency. In particular, the four exempla illustrated in Figs 4 and 5 are quite representative of most observed cases. The first stable patient in Fig 4 (blue continuous line with open circles vs dashed red line with asterisks) exhibits a constant improvement in tapping frequency (up to about 190 taps/min) during the four hours post-dosing. The second patient in Fig 4 (green continuous line with triangles vs. dashed magenta line with crosses) is still quite stable, but he/she exhibits a smaller initial increment in tapping frequency (to about 150 taps/min) and a moderate decline during the third and fourth hours. The first fluctuating patient in Fig 5 (blue continuous line with open circles vs dashed red line with asterisks) exhibits just a small increase in tapping frequency (only to about 130 taps/min) which is maintained until the third

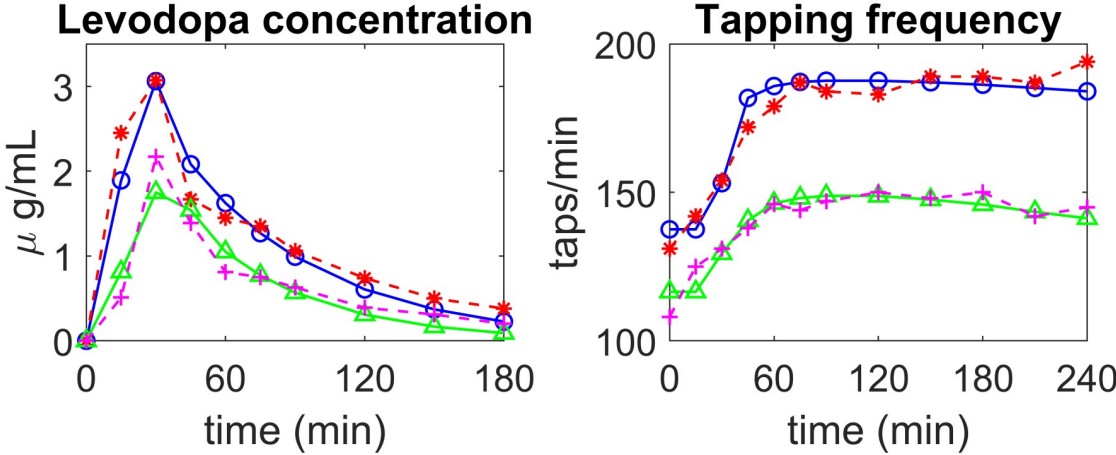

**Fig 4. Examples of model fitting to real data in two patients of the *first* group (stable).** The left panels show LD concentration in plasma, the right panels the finger tapping rate. *Patient 5*—experimental data: dashed red lines with asterisks; simulation results: continuous blue lines with open circles; *Patient 6*—experimental data: magenta dashed lines with crosses; simulation results: continuous green lines with triangles.

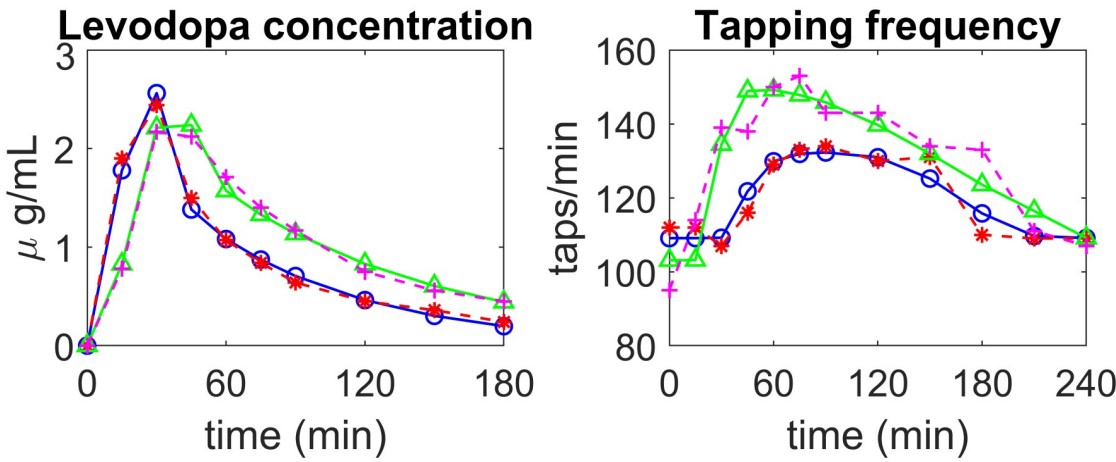

**Fig 5. Examples of model fitting to real data in two patients of the _second_ group (fluctuating).** The left panels show LD concentration in plasma, the right panels the finger tapping rate. _Patient 2_—experimental data: dashed red lines with asterisks; simulation results: continuous blue lines with open circles; _Patient 12_—experimental data: magenta dashed lines with crosses; simulation results: continuous green lines with triangles.

hour; at his time, tapping frequency exhibits a quick fall to the initial level. Finally, the second fluctuating patient in Fig 5 (green continuous line with triangles vs. dashed magenta line with crosses) displays an intermediate behaviour between that of the previous two illustrated patients. He/she exhibits an increase up to 150 taps/min and quite a rapid continuous decline from the first hour afterwards, with a return to baseline value within the four hours of monitoring. As the figures show, the model, with suitable parameter values, can mimic all these different typologies quite well.

The complete list of figures (13 patients of the first group and 13 patients of the second) can be found in the S2 Material.

In order to quantify the adequacy of the fitting procedures, the $R^2$ values for plasma LD concentration and for tapping frequency, in all patients, are reported in Table 3. $R^2$ represents the quotient of the variance explained by the model to the total variance, and provides a measure of how the model is able to replicate the observed outcomes. Results in Table 3 indicate that the model with optimal parameter values is able to provide an adequate reproduction of the available data.

For what concerns plasma LD concentration, we can observe that just in one case (patient 8 of group 1) $R^2$ was quite low (below 0.75). In this case, however, a better fitting may be achieved by modifying the initial guess (we used just a single initial guess for all plasma concentration fitting procedures). For what concerns the tapping frequency, fitting is adequate for all patients of group 1 ($R^2 > 0.8$) with the only exception of patient 2. However, fitting of tapping frequency is more difficult for patients of the second group. In three cases (patients 3, 6 and 9) the $R^2$ was lower than 0.75. In case of patient 9 of group 2, this can be ascribed to the presence of a remarkable outlier (that we did not remove from fitting). In the other cases (patient 2 of group 1 and patients 3 and 6 of group 2) the problem may derive from the very fast increase in tapping rate occurring during the first 30 minutes from LD administration, which the model fatigues to reproduce (see S2 Material).

To better illustrate differences between the two groups, Fig 6 shows boxplots of all the estimated parameters for the two groups, together with the p value obtained from the Wilcoxon signed-rank test. For what concerns the pharmacokinetics parameters (upper line in Fig 6) no significant statistical differences can be observed. Thus, we can conclude that differences

**Table 3. Goodness of fitting through $R^2$ evaluation.**

| First group (stable) | | |
|---|---|---|
| Patient | $R^2$_LD | $R^2$_tapping |
| 1 | 0.9674 | 0.9129 |
| 2 | 0.9898 | 0.7411 |
| 3 | 0.9942 | 0.8747 |
| 4 | 0.9846 | 0.8853 |
| 5 | 0.9287 | 0.9242 |
| 6 | 0.8932 | 0.8794 |
| 7 | 0.9846 | 0.9198 |
| 8 | 0.7018 | 0.8414 |
| 9 | 0.9950 | 0.9250 |
| 10 | 0.9331 | 0.8714 |
| 11 | 0.9874 | 0.9328 |
| 12 | 0.8955 | 0.8532 |
| 13 | 0.9616 | 0.8914 |
| **mean** | **0.9398** | **0.8810** |
| Second group (fluctuating) | | |
| 1 | 0.9323 | 0.8366 |
| 2 | 0.9902 | 0.9026 |
| 3 | 0.9893 | 0.6713 |
| 4 | 0.9829 | 0.8713 |
| 5 | 0.9938 | 0.8316 |
| 6 | 0.9907 | 0.7148 |
| 7 | 0.9010 | 0.9066 |
| 8 | 0.9104 | 0.8193 |
| 9 | 0.8870 | 0.7082 |
| 10 | 0.9659 | 0.7694 |
| 11 | 0.9883 | 0.7995 |
| 12 | 0.9893 | 0.8695 |
| 13 | 0.9159 | 0.7735 |
| **mean** | **0.9567** | **0.8057** |

between the two groups cannot be ascribed to LD plasma kinetics. This result agrees with the conclusion of former studies [8,21,42].

Conversely, significant differences between the two groups, overcoming the Bonferroni correction, can be observed looking at some parameters of the pharmacodynamics (second and third line in Fig 6): in particular, significant differences concern parameter $k_{e3}$, which represents the drug removal rate from the effect compartment (p = 0.0011) and $N_D$ which establishes the slope of the Hill curve (p = 0.0009). In particular, these parameters are significantly higher in the fluctuating patients, denoting a more rapid drug removal from the brain and a more rapid decay of the drug effect (the higher $N_D$, the higher the slope of the Hill curve from saturation to no effect). It is worth noting that other three parameters exhibit some differences, but which do not reach statistical significance.

A significant limitation of the present study, which requires further analysis, consists in having maintained a single set of parameters in the neurocomputational model for all patients, that is all differences in the observed responses are ascribed to differences in the LD pharmacodynamics. In order to investigate whether changes in the BG can affect the robustness of parameter estimates, we repeated some of the previous simulations (with reference to a typical

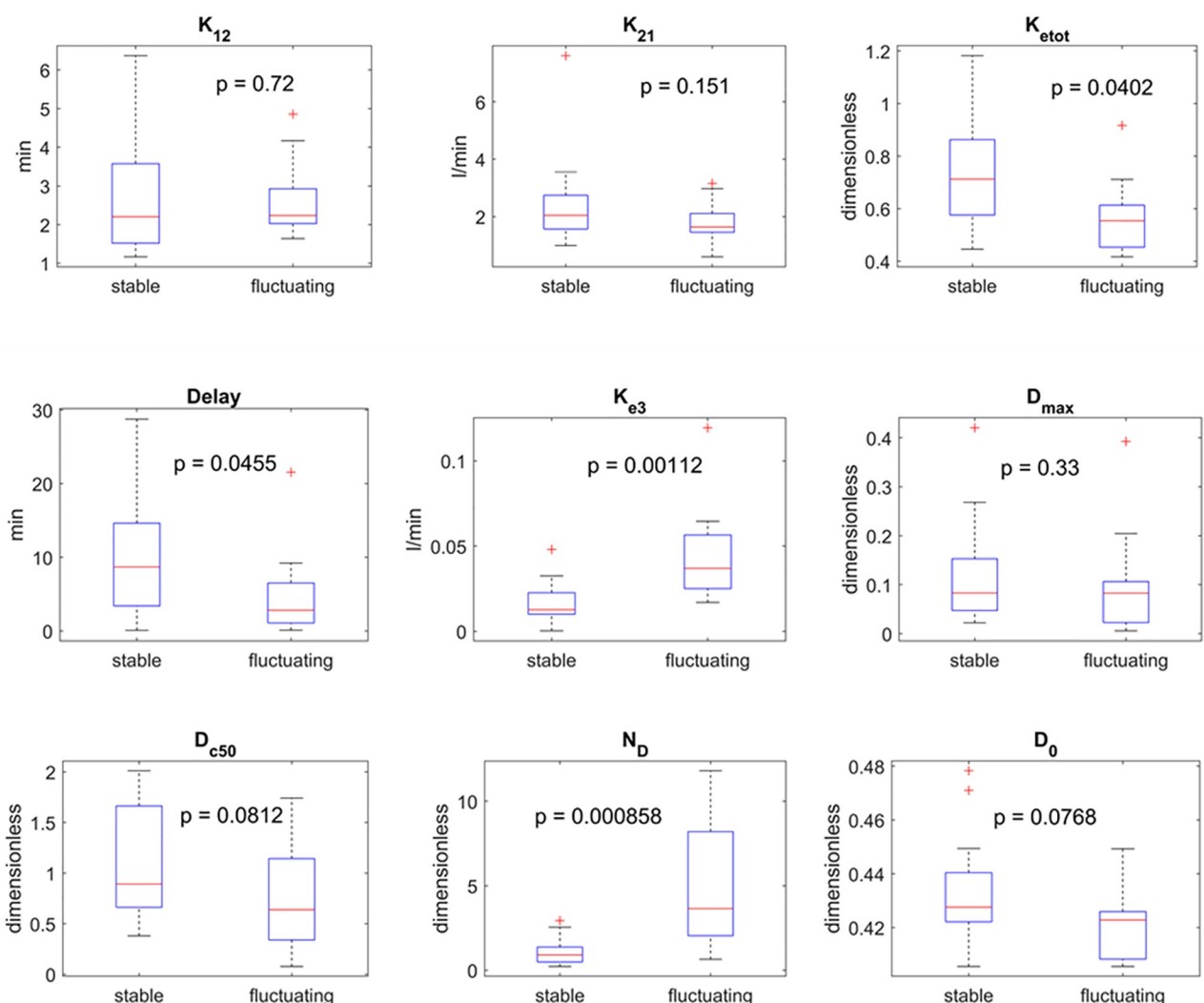

**Fig 6. Comparison between the estimated parameters in the two classes of patients (see Table 1 for the meaning of symbols).** The panels show the boxplot for each parameter in each class (minimum, first quartile, median, third quartile, maximum and possible outliers). The p value obtained via a Wilcoxon rank sum test are also displayed in each panel, without Bonferroni correction. The upper row refers to the pharmacokinetics parameters, the second and third rows to the pharmacodynamics. If a Bonferroni correction is applied (i.e., the p value is multiplied by nine), differences in parameters $k_{e3}$ and $N$ remain highly significant between the two classes.

patient of group 1 and to a typical patient of group 2) by modifying the value of the synapses in the neural network. To this end, we used the pattern of synapses obtained in a previous work starting from a completely untrained network, at different levels of training (see Fig 3 in [22]). In particular, synapses after 100 epochs are representative of a moderate skill, and can be considered typical of a PD patient. These are the synapses used throughout the previous simulations. To simulate different conditions, we also used synapses obtained after 60, 80, 150 and 200 epochs. The first are representative of poor skill, the last of very high skill. Results are shown in the S3 Material. Briefly, if all other parameters of pharmacodynamics are maintained at the estimated level, a change in the synapses causes a remarkable change in basal tapping rate (i.e., the rate before LD administration; of course, a higher skill corresponds to a higher rate). Hence, we empirically modified the value of parameter $D_0$, which represents the dopaminergic input before the treatment, as a function of the synapse level, so as to re-obtain

approximately the same initial tapping rate as in the patient. In these conditions, we observed that the temporal pattern of the tapping rate during the treatment is almost unaffected by alterations in the synapse strength, provided the other pharmacodynamics parameters are maintained at the value estimated before. In particular, we cannot ascribe the differences between stable and fluctuating patients to changes in the synapse values. In conclusion, we claim that the only parameter substantially affected by the synapses is $D_0$, whereas estimation of the other parameters is quite robust.

## Simulation of learning in two exemplary patients

The model, with parameter assigned, can potentially be used to explore the behavior of a specific patient in various conditions, under LD treatment. In particular, as pointed out by Beeler et al. [14] and confirmed by computer simulations in our previous work [22], patient performance (in particular, the basal tapping rate) can be strongly affected by learning, via a modification of the synapse strength in the BG. Rewards may improve patient symptoms, whereas punishments may further accentuate bradykinesia. Learning, in turn, is significantly modulated by dopamine, and so a patient can exhibit a different response to training depending on the moment when training is performed during the treatment and on the stability of his/her response to LD. Due to the potential great impact in clinics, it may be of value to point out these possible differences in the two groups via computer simulations. Two examples are given below, concerning a stable and a fluctuating patient, respectively. In each example we assume that the simulated patient is subjected to a short training phase, performed as described in detail in a previous paper [22]. Briefly, the training period consists in 50 epochs: During each epoch, the patient alternatively receives one of the two commands (corresponding to the two possible finger actions) but in the presence of much external noise, to make the action choice difficult, so that the patient can learn from errors. Immediately after the selection of a choice, a reward or punishment is given via a phasic change in dopamine (transiently doubling or nulling the current D value, respectively). Training is performed either during the first hour after LD administration, when the drug effect is maximal, or after several hours, when its effect is declining.

Results, shown in Fig 7, are just indicative since the outcomes of the training procedure vary randomly due to the presence of noise. However, the results provide general, important suggestions. In particular, as illustrated in these particular exempla, training, if performed at one hour after LD administration, may have a mild beneficial effect (resulting in a higher tapping frequency) both in the stable patient and in the fluctuating patient. Although in some cases this effect can be absent, due to the effect of noise, in any case training does not result in a poor outcome. Conversely, if performed several hours after LD administration, training results in a clear slowing down of the tapping rate, which is moderate in the patient of the first group (after training, the tapping frequency still remains elevated compared with the basal pre-LD condition: indeed, LD is still efficacious in this group) but has extremely negative effect in the fluctuating patient (tapping frequency abruptly decreases even below the basal pre-LD value). The latter result points out the danger of aberrant learning, especially in patients of the second groups at hours from medication.

## 4. Discussion

The present work has several objectives. First, we wished to further validate an integrate (pharmacodynamical + neurocomputational) model developed in recent years [20–22] on a wider patient population. Second, we wished to implement and test an automatic procedure for

### Patient 5 group 1 (stable)

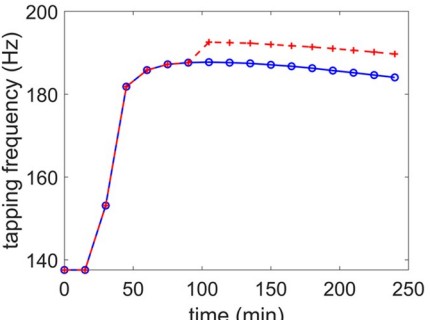 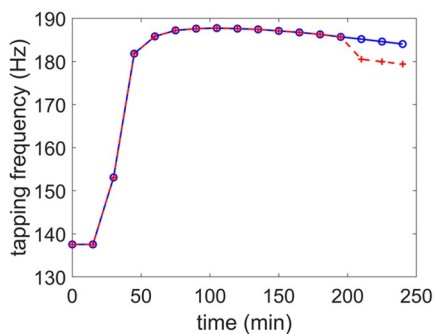

### Patient 5 group 2 (fluctuating)

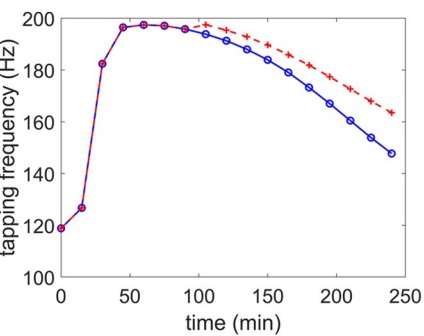 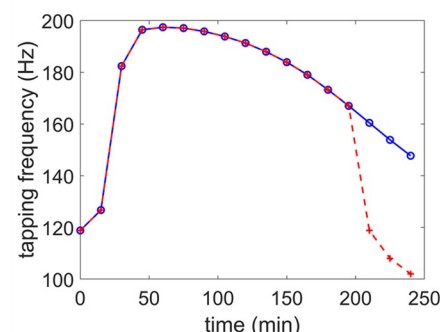

**Fig 7. Examples of the effect of training in two exemplary patients, at different moments after LD administration.** Continues blue lines with open circles represent the temporal pattern of tapping frequency without training (i.e., the curves estimated on the patient); the dashed red lines with crosses represent the finger tapping frequency after training. The upper panels refer to a patient (n. 5) of the first group, whereas bottom panels to a patient (n. 5) of the second group. Training was performed either at 90 min after LD administration, when its effect was maximal (left panels) or at 210 min after administration (right panels), when the effect was vanishing. Training consisted in 50 epochs of finger response to a randomized input, with much superimposed noise. Striatal synapses where trained with a Hebb rule via reward or punishment signals (peaks or drops of dopaminergic inputs, respectively) as described in [22]. It is worth noting that training may result in a slight motor improvement if performed after 1 hour, especially useful in patients of group 2. Conversely, training has a detrimental effect if performed after hours, and this damage is especially dramatic in patients of the second group.

parameter fitting, which allows the model to be automatically adjusted on the individual patient. Third, we looked for the clinical significance of the estimated parameter values.

These objectives have been adequately achieved.

In all examined cases, the model was able to simulate the temporal patterns of LD plasma concentration and of finger tapping frequency pretty well, during several hours after drug administration. The adequacy of fitting is documented by the $R^2$ values reported in Table 3.

Furthermore, the estimation procedure sheds some light into the possible reasons for the observed differences between stable and fluctuating patients. By comparing the estimated parameter values in the two classes, no significant differences emerged for what concerns LD plasma kinetics. This result agrees with previous data [8,42], and also agrees with the observations summarized in Contin et al. [3], showing that the pharmacokinetic parameters remain substantially unchanged with disease progression. Conversely, clear differences are evident for

what concerns the pharmacodynamic parameters estimated on the individual patient. Actually, two parameters are significantly different in the two classes: the rate of LD removal from the effect compartment ($k_{e3}$) and the Hill coefficient ($N_D$). The latter determines the slope of LD effect relationship on the striatal neurons. These two differences enlighten why fluctuating patients have a much more unstable behaviour: first, due to the higher values of $k_{e3}$, drug concentration more rapidly declines in the effect compartment after drug administration; second, due to the high value of $N_D$, even a small decrease in drug concentration can induce a dramatic fall in neuron response. These two factors are able to explain "per se" why patients in the fluctuating class exhibit an abrupt deterioration in the velocity of finger movement (hence bradykinesia) just a few hours after the beginning of treatment.

However, other parameters in pharmacodynamics exhibit interesting differences between the two classes, in particular, the delay of drug action on striatal neuron ($T$), and the concentration at 50% effect ($D_{c50}$); however, they do not reach a statistical significance (especially after Bonferroni correction).

In line with previous observations [8,42], the main biological consideration that can be drawn from the present results is that differences in LD response between stable and fluctuating PD patients could be ascribed above all to altered drug cerebral kinetics. In particular, the higher values of $k_{e3}$ found in the fluctuating, more severely affected group may reflect an increased rate of synthesis and release of LD-derived dopamine by the surviving nigrostriatal projections, up to a potential conversion of LD outside dopaminergic neurons [43]. According to this hypothesis, the severity of PD symptoms, which reflects nigral cell loss, was previously reported to significantly correlate with $k_{e3}$ [8]. Moreover, an unstable response to oral LD is associated with an increased value of the sigmoidicity index $N_D$, a parameter which might indicate the degree of drug receptor occupancy [42]. This could mean that in stable patients the response to LD dosing is gradual in onset, graded and sustained in magnitude even when a small fraction of receptors is occupied by the drug. On the contrary, in fluctuating patients LD response often appears to be "all or nothing", more rapid in the onset and abrupt in the offset, matched to the occupancy of the majority of receptors.

Results of our fitting procedure support the results of previous studies, which used compartmental models to directly mimic the dynamical relationship between LD administration and finger tapping frequency. In particular, Contin et al. [8,39,42] observed that patients with wearing-off phenomena required about two-fold levels of LD concentration compared with stable patients. However, some dissimilarities are evident in the meaning of estimated parameters between our model and Contin et al. Both in our study and in the studies by Contin et al., differences in pharmacokinetic parameters are not relevant. Indeed, as evident in Table 3 in Contin and Martinelli [3], LD plasma kinetics is not significantly affected by the disease progression. For what concerns pharmacodynamics, as in our study Contin et al. observed no significant differences in the maximum effect (equivalent to our parameter $D_{max}$) between the two classes, a significant increase in the Hill coefficient, $N_D$, for the fluctuating group, and an increase in $k_{e3}$. At odd with our model, however, Contin et al. observed a significant increase in concentration producing 50% of maximum effect with disease progression, a parameter which is not significantly different in our estimations or even reduced in fluctuating patients.

However, it is important to stress that our approach is different from the purely compartmental approach used in previous studies; indeed, in our approach the two-compartment model is used to provide the dopaminergic input to a neuro-computational model of the BG, which, in turn, produces the finger tapping response as a continuous balance between the Go and NoGo pathways. An increase in the output of the effect compartment excites the winning Go Neurons, and depresses the NoGo neurons, thus favouring a rapid selection of the correct action. Accordingly, the values of parameters in our model, and in part their meaning, are

different from the corresponding parameters in previous studies, where the output of the effect compartment was directly fitted to the Finger Tapping frequency.

An important limitation of our approach is that all parameters in the BG model were maintained at the same value, independently of the particular patient. The fundamental reason for this restriction is that we do not have enough data available, to attempt a complete parameter estimation which incorporates both the pharmacodynamics and the neural parameters. This limitation should be overcome in future work.

Indeed, some authors [11] proposed that the compromised response of PD patients (especially bradykinesia) compared with normal subjects is the result of two concurrent factors: a "gain factor", which basically reflects the level of dopamine (or LD effect) on the striatal neurons, and a "learning factor", which reflects previous cortico-striatal synaptic plasticity. While the first is directly affected by the drug, the second reflects a previous history of rewards and punishments in the subject and, more important, is influenced by the scarce capacity of PD patients to learn from experience as a consequence of a depressed dopamine pool. In a previous study ([22], we confirmed this point of view, showing, with the use of a BG neurocomputational model, that the finger tapping response is modulated by both factors, and that PD patients reliably have less developed cortico-striatal synapses besides lower dopamine levels. This poor synapse development is reflected in a less evident differentiation between the Go and NoGo pathways compared with that occurring in healthy subjects.

To account for this phenomenon, in the present study we used values of cortico-striatal synapses from our previous work (see Fig 3 in [22]), to reflect poor previous learning. Of course, it is probable that the values of these synapses are different from one patient to the next, reflecting his/her individual past history. However, to fit all these parameters (pharmacodynamics + synapse values) more data should be acquired in future studies, besides those of the finger tapping response, to avoid overfitting. It is well known, in fact, that estimation of an excessive number of parameters, compared with the complexity of the data available, results in poor accuracy of the parameter estimates and, above all, in a poor model capacity to generalize, i.e., in a negative impact of new data on model performance [40].

Nevertheless, as described in section Results (but see also S3 Material) in order to provide a first glance on this problem, we repeated the simulations on two representative patients by modifying the synapse level in the BG. Results suggest that only the estimation of parameter $D_0$ is significantly affected by the synaptic weight, whereas the other estimates appear quite robust. In particular, it does not seem possible to differentiate between stable and fluctuating patients on the basis of the synapse value in the striatum (Go and NoGo): patient 1 in the S3 Material always exhibits the typical behaviour of a stable patient, while patient 2 always exhibits instability despite different synaptic strengths. Of course, we did not perform a complete exam of the problem, since other parameters in the neural network may produce different results, but our analysis provides a preliminary well-founded indication on the robustness of our conclusions.

Of course, the main advantage of the present approach, compared with a purely compartmental model, is that new predictions can be built on patient behaviour, on the basis of the neurocomputational simulation. As an example of these possibilities, we illustrated the effect of a brief training procedure, performed on two patients at different hours after LD administration. Results show that fluctuating patients can exhibit a different behaviour, depending on whether training is performed in the early phase after LD administration, or is performed a few hours later. In the first moment, training may have a significant benefit, resulting in a prolongation of the LD effect, via a synapse reinforcement. In the second moment, learning may be deleterious, causing even a stronger bradykinesia via synapse weakening. These differences are manifest even in stable patients, but much less dramatic.

This peculiar effect of training is a consequence of the Hebb rule for synaptic potentiation or depotentiation adopted in our model [20]. In this rule, the activity of the post-synaptic neurons (either in the Go or in the NoGo pathways) is compared with a threshold, and the synapse is potentiated if the post-synaptic activity is above threshold, but depressed in case of weak post-synaptic activity. This mechanism explains why learning is strictly related with the dopaminergic input to the BG, and why training a PD patient may be problematic due to the scarce benefit of rewards, caused by an insufficient dopamine peak, and by a stronger effect of punishments, caused by excessive dopamine dips. This represents just a first step toward a more accurate, quantitative understanding of a complex chain of events, which incorporate the dopamine level, the synapse plasticity and the neuron active response.

## Supporting information

**S1 Material.**
(DOCX)

**S2 Material.**
(DOCX)

**S3 Material.**
(DOCX)

## Author Contributions

**Conceptualization:** Mauro Ursino.

**Data curation:** Giovanna Lopane, Giovanna Calandra-Buonaura, Pietro Cortelli, Manuela Contin.

**Formal analysis:** Mauro Ursino, Elisa Magosso.

**Methodology:** Mauro Ursino, Elisa Magosso.

**Software:** Mauro Ursino.

**Supervision:** Mauro Ursino, Manuela Contin.

**Validation:** Mauro Ursino, Elisa Magosso.

**Visualization:** Elisa Magosso, Manuela Contin.

**Writing – original draft:** Mauro Ursino.

**Writing – review & editing:** Elisa Magosso, Giovanna Lopane, Giovanna Calandra-Buonaura, Pietro Cortelli, Manuela Contin.

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
