## [Decision Letter · Decision Letter 0]

24 Dec 2019

PONE-D-19-25770

Mathematical Modeling and Parameter Estimation of Levodopa Motor Response in Patients with Parkinson Disease

PLOS ONE

Dear Prof. Ursino,

Thank you for submitting your manuscript to PLOS ONE. After careful consideration, we feel that it has merit but does not fully meet PLOS ONE’s publication criteria as it currently stands. Therefore, we invite you to submit a revised version of the manuscript that addresses the points raised during the review process.

The manuscript should be revised for clarity. The design of the study should be explained more clearly. The model and data analysis should be described in detail.

Three-compartmental model - what are the compartments? The authors need to explicitly state what the compartments represent up-front so the readers will have the right context in mind.

Although Hill laws are widely used in pharmacodynamics, the authors need to explain how Hill law can "mimic how LD can affect striatal neurons" ; how are the neurons modeled? What is the interaction between the Hill law and the modeled neurons?

** Methods

Please, explain the basis for the network architecture (topological structure) and provide relevant sources from the literature justifying the model.

** Figures

Some figures could be combined together. For example in Figures 4 and 5 the left 2 and right two panels could be combined together using different symbols. Similarly Figure6A and Figure6B could be combined together.

** Results

"Results of the fitting procedure" Please, explain that there are two different patient groups used to fit the model on, and what the differences between the groups are, and then how you fit the model to each group.

Although some details are provided in the figure legends the results section description of figures 4 and 5 are not described in sufficient detail in the Results. Minimally, the authors should describe in detail the answers to the following questions in the Results: What are the red and blue traces in figures 4 and 5? What are the differences between the patients? Why is it important to show both of them?

The authors state that "R^2 values for plasma...in Table 3" . Why is this important? What does

the table show? Please describe at least minimally.

"Simulation of learning in two exemplary patients" ... The authors state, "Although this subject

is well beyond the aim of the present study, two examples are given below, concerning a stable

and a fluctuating patient," . If this subject is beyond the aim of the study, why do the authors

present results on the subject? There must be some benefit to describing this result, so the

authors should describe their work more clearly, why it was done, and what limitations

it has instead of stating it's beyond the scope of the paper but then describing it anyway.

As mentioned above, the authors should clearly define the stable and fluctuating patient groups.

** Discussion

Some parts of discussion could be used for elaboration of Results

Please, clarify what biological conclusions they can reach with their mathematical model. Clarify whether certain parameters in the compartmental models were significantly different between the two patient groups.

How robust is this biological conclusion? To reach this conclusion, each patient had the same parameters for their neurocomputational model, but different parameters for their compartment model. If they instead used the same parameters for their compartmental model for each patient, and then chose different parameters in the neurocomputational, could they still fit the data? If so, it would seem that their biological conclusion based on their mathematical model is not so robust.

Please, run a spell-checker:

tipically -> typically

We would appreciate receiving your revised manuscript by Feb 07 2020 11:59PM. To enhance the reproducibility of your results, we recommend that if applicable you deposit your laboratory protocols in protocols.io, where a protocol can be assigned its own identifier (DOI) such that it can be cited independently in the future. For instructions see: http://journals.plos.org/plosone/s/submission-guidelines#loc-laboratory-protocols

We look forward to receiving your revised manuscript.

Kind regards,

Gennady Cymbalyuk, Ph.D.

Academic Editor

PLOS ONE

Journal Requirements:

2. Thank you for inlcuding your ethics statement; "PD patients were recruited within the "Bologna Motor and Nonmotor Prospective Study on Parkinsonism at Onset” (BoProPark study, protocol number 09070, local ethics committee of Bologna). Kinetic-dynamic modelling of LD motor response was part of the study. Patients gave their written informed consent to participate in the study and to publish the data."

3. Please ensure that you refer to Figure 2 in your text as, if accepted, production will need this reference to link the reader to the figure.

Reviewers' comments:

Reviewer's Responses to Questions

**Comments to the Author**

1. Is the manuscript technically sound, and do the data support the conclusions?

Reviewer #1: Partly

Reviewer #2: Yes

2. Has the statistical analysis been performed appropriately and rigorously? 

Reviewer #1: Yes

Reviewer #2: I Don't Know

3. Have the authors made all data underlying the findings in their manuscript fully available?

Reviewer #1: No

Reviewer #2: Yes

4. Is the manuscript presented in an intelligible fashion and written in standard English?

Reviewer #1: No

Reviewer #2: Yes

5. Review Comments to the Author

Reviewer #1: The authors present a study on modeling pharmacodynamics of levodopa administration and its

impact on behavior through a computational network model of basal ganglia and other brain

areas, that then produce behavioral outputs. The work integrates clinical and modeling

data to better understand dopamine/levodopa role in behavior for specific patient populations.

Although the model is potentially useful and the subject of the work interesting, the writing

and presentation of the results need considerable re-writing/clarifications in order for

readers and reviewers to understand whether the results are worth reporting. The model source

code also needs to be shared on a public website so others can replicate the results.

Detailed comments on individual manuscript components follow.

** Introduction

The introduction needs to be revised to introduce the model and its components

clearly. right now the main components are not mentioned, and how they interact.

authors should run a spell-checker:

tipically -> typically

three-compartmental model - what are the compartments? in computational

neuroscience this can mean several different things, so the authors

need to explicitly state what the compartments represent up-front so the

readers will have the right context in mind.

Although Hill laws are widely used in pharmacodynamics, the authors need to explain how

Hill law can "mimic how LD can affect striatal neurons" ; how are the neurons modeled?

what is the interaction between the Hill law and the modeled neurons?

** Methods

Can the authors explain the basis for their network architecture (what they refer

to as topological structure) and relevant sources from the literature for

their decisions

The authors should share the model source code on a public accessible webpage such

as github.com or modeldb.

** Figures

Some figures could be combined together. For example in Figures 4 and 5 the left 2

and right two panels could be combined together using different symbols.

Similarly Figure6A and Figure6B could be combined together.

** Results

"Results of the fitting procedure" In this section the authors should first introduce

that there are two different patient groups that they intend to fit the model on, and

what the differences between the groups are, and then how they fit the model to each

group. Otherwise, it is confusing for the reader.

Although some details are provided in the figure legends the results section description

of figures 4 and 5 are not described in sufficient detail in the Results. Minimally, the authors

should describe in detail the answers to the following questions in the Results: What are the red and blue

traces in figures 4 and 5? What are the differences between the patients? Why is it important

to show both of them?

The authors state that "R^2 values for plasma...in Table 3" . Why is this important? What does

the table show? Please describe at least minimally.

"Simulation of learning in two exemplary patients" ... The authors state, "Although this subject

is well beyond the aim of the present study, two examples are given below, concerning a stable

and a fluctuating patient," . If this subject is beyond the aim of the study, why do the authors

present results on the subject? There must be some benefit to describing this result, so the

authors should describe their work more clearly, why it was done, and what limitations

it has instead of stating it's beyond the scope of the paper but then describing it anyway.

As mentioned above, the authors should clearly define the stable and fluctuating patient groups.

** Discussion

Some parts of discussion could be used for elaboration of Results

Reviewer #2: In this paper, the authors develops and uses a mathematical model to understand certain aspects of Parkinson's disease (PD) and levodopa (LD) treatment. The model is fit to and compared with the clinical results on 26 patients with PD who were undergoing LD treatment and a finger tapping test which assesses the motor symptoms of PD and the effect of any LD therapy. The clinical results are (i) LD plasma concentrations at a sequence of time points for each patient following LD treatment and (ii) patient performance on finger tapping tests. The mathematical model consists of a so-called compartmental model (differential equations describing the concentration and flow of LD in a patient's body) and a neurocomputational model (differential equations describing the activity of neurons) which ultimately output the tapping frequency for a patient. The authors developed an automated procedure to choose parameters in their model to fit individual patient data.

The authors declare 3 aims for this work. (1) Further validate that integrated mathematical model (a compartment model integrated with a neurocomputational model) on a larger patient set. (2) Implement and test their automated procedure to choose parameters to fit individual patients. (3) See if there is a significant differences in estimated parameters between patients with a stable versus a fluctuating response to LD therapy. The authors claim to accomplish these 3 aims and I agree.

In my view, this is interesting work that is worthy of publication. Some novel and important aspects include

-an integrated compartment/neurocomputational model,

-the use of clinical data to tune parameters to individual patients,

-the discovery that patients in the two groups (fluctuating versus stable response to LD therapy) have significantly different parameters in their compartment models.

I have a few comments and suggestions.

-I think the authors should more clearly state what biological conclusions they can reach with their mathematical model. This appears to be that certain parameters in the compartmental models were significantly different between the two patient groups.

-Further, how robust is this biological conclusion? To reach this conclusion, each patient had the same parameters for their neurocomputational model, but different parameters for their compartment model. If they instead used the same parameters for their compartmental model for each patient, and then chose different parameters in the neurocomputational, could they still fit the data? If so, it would seem that their biological conclusion based on their mathematical model is not so robust.

6. PLOS authors have the option to publish the peer review history of their article (what does this mean?). If published, this will include your full peer review and any attached files.

Reviewer #1: No

Reviewer #2: No

---

## [Author Response · Author response to Decision Letter 0]

1 Feb 2020

All responses can be found in the enclose "Response to The Reviewvers" file.

The content is copied below again:

PONE-D-19-25770

Mathematical modeling and parameter estimation of levodopa motor response in patients with parkinson disease

PLOS ONE

Reply to the Editor

Thank you for managing the paper and the important suggestions.

All main suggestions raised by the Reviewers have been addressed as closely as possible, as illustrated in the comments below. We also addressed these additional Journal requirements:

1) The file naming has been checked

2) The current ethics statement has been amended to include the full name of the ethics committee (page 10 lines 1-3),

3) We now refer to Fig. 2 in the manuscript at page 13 line 5. 

Finally, the laboratory protocol has been deposited in protocols.io, as requested 

(http://dx.doi.org/10.17504/protocols.io.bbq8imzw). 

 

Reply to the Reviewer 1

We thank the Reviewer for the interesting comments and advice, which significantly contribute to the improvement of the present work.

** Introduction – 

The introduction needs to be revised to introduce the model and its components clearly. Right now the main components are not mentioned, and how they interact. Three-compartmental model - what are the compartments? in computationa lneuroscience this can mean several different things, so the authors need to explicitly state what the compartments represent up-front so the readers will have the right context in mind. Although Hill laws are widely used in pharmacodynamics, the authors need to explain how Hill law can "mimic how LD can affect striatal neurons" ; how are the neurons modeled? what is the interaction between the Hill law and the modeled neurons?

We included some explanations in the Introduction, to clarify the main components of the model and their interactions. In particular, we describe how neurons are modelled, what is the meaning of the three compartments, and how the Hill law is used to link LD with the neuron response (from page 5 line 21 to page 6 line 12 of the Introduction). 

Authors should run a spell-checker: tipically -> typically

We ran a spell checker to correct all possible typographical errors. Thank you!

** Methods 

Can the authors explain the basis for their network architecture (what they refer to as topological structure) and relevant sources from the literature for their decisions

A description of the structure of the network is now provided from page 8 line 15 to page 9 line 6 of the Method section, where the most relevant references from the literature are also reported (page 8, line 16). We also included a new panel in Fig. 1 (panel b) to summarize the architecture of the main neural pathways in the BG model. 

The authors should share the model source code on a public accessible webpage such as github.com or modeldb.

The source code, with comments, has been inserted into the ModeldB repository at the following link http://modeldb.yale.edu/261624 . 

A short section named” Data Availability” has been included (page 24 lines 1-8).

** Figures 

Some figures could be combined together. For example in Figures 4 and 5 the left 2 and right two panels could be combined together using different symbols. Similarly Figure6A and Figure6B could be combined together.

As requested, we combined the left two and the right two panels in Figs. 4 and 5 together, using different colours and different symbols. Moreover, as suggested, we combined Figs 6a and 6b together into a single figure. Thank you!

** Results 

"Results of the fitting procedure" Please, explain that there are two different patient groups used to fit the model on, and what the differences between the groups are, and then how you fit the model to each group.

We included a brief explanation on the two patient groups at page 14 lines 11-17 in section Results, as requested. Moreover, we explicitly confirm here that we used the same fitting procedure (as described in Method section) to fit each patient in each group, without differences between the two procedures. Hence, differences between the two groups are merely a consequence of the final parameter estimate, not of model structure or of the fitting procedure. 

Although some details are provided in the figure legends the results section description of figures 4 and 5 are not described in sufficient detail in the Results. Minimally, the authors should describe in detail the answers to the following questions in the Results: What are the red and blue traces in figures 4 and 5? What are the differences between the patients? Why is it important to show both of them?

We added all requested information at from page 14 line 18 to page 15 line 14 of section Results. We now explain the differences between the red and blue lines, and the differences among the four patients illustrated. Indeed, each patient is representative of a different behaviour: we show two typical but slight different pieces of behaviour for stable patients in Fig. 4, and the patterns of two fluctuating patients with different form of instability in Fig. 5. 

The authors state that "R^2 values for plasma...in Table 3" . Why is this important? What does the table show? Please describe at least minimally.

The significance of R2 and its importance to quantify the quality of fitting are now described from page 15 lines 18 to page 16 line 6 of section Results. In the same paragraph we now comment on the significance of the results in Tab. III. Thank you. 

"Simulation of learning in two exemplary patients" ... The authors state, "Although this subject is well beyond the aim of the present study, two examples are given below, concerning a stable and a fluctuating patient," . If this subject is beyond the aim of the study, why do the authors present results on the subject? There must be some benefit to describing this result, so the authors should describe their work more clearly, why it was done, and what limitations it has instead of stating it's beyond the scope of the paper but then describing it anyway.As mentioned above, the authors should clearly define the stable and fluctuating patient groups.

We agree with this important comment of the Reviewer. Hence, we eliminated the previous sentence and we modified the text, by including a description on why it is important to analyse learning during LD treatment in individual patients (from page 17 line 20 to page 18 line 2). 

Some parts of discussion could be used for elaboration of Results

We modified the Discussion, by moving some portions to section Results, as suggested by the Reviewer and how it has been described in the previous responses. In particular, all comments on the values of R2, to evaluate the accuracy of fitting, have been moved from the Discussion to the Results (from page 15 line 22 to page 16 line 6) in section Results. Thank you!

Reply to the Reviewer 2

We thank the Reviewer for having appreciated our work and for the interesting comments and advice, which significantly contribute to the improvement of the present study.

- I think the authors should more clearly state what biological conclusions they can reach with their mathematical model. This appears to be that certain parameters in the compartmental models were significantly different between the two patient groups.

In order to address this important comment of the Reviewer, we introduced a new paragraph in section Discussion (page 20 lines 5-17) to comment on the biological meaning of the obtained results, with particular emphasis on the two parameters (the Hill coefficient and the drug removal coefficient) which are significantly different between the two classes. 

- Further, how robust is this biological conclusion? To reach this conclusion, each patient had the same parameters for their neurocomputational model, but different parameters for their compartment model. If they instead used the same parameters for their compartmental model for each patient, and then chose different parameters in the neurocomputational, could they still fit the data? If so, it would seem that their biological conclusion based on their mathematical model is not so robust.

We completely agree with this important concern of the Reviewer. To address it, we performed some new simulations on two representative patients (patient 1 of the stable group and patient 2 of the fluctuating group) by modifying the synapse values in the BG neural model, still maintaining the same parameters for LD pharmacodynamics previously estimated. In order to choose new synapses, we used the pattern of synapses obtained in a previous work (see Fig. 3 in Ursino and Baston, Europ. J. Neurosci, 2018) at different levels of training (these synapses are representative of poor skill levels, intermediate skill levels and optimal skill levels). The aim of these simulations was to understand whether different synapses significantly modify the pattern of the finger tapping response. Results, presented in Supplementary material III, suggest that a change in the synapse strength is reflected in a significant change in the baseline finger tapping rate (i.e., the rate before LD administration). Hence, to reproduce the same patient’s behaviour, parameter D0 (which represents the initial level of the dopaminergic term) must be modified. We then repeated the simulations, by using an appropriate value of D0 for each synapse level, to set quite a correct initial tap rate, and observed that, in these conditions, changes in the synapse strength do not affect the behaviour. Of course, this is not a complete exam of the problem, since other parameters in the network, or a different distribution of Go vs. NoGo synapses may produce different results, but it provides a strong indication that, with the only exception of parameter D0, the other pharmacodynamics parameters are quite robust. These aspects are now commented in a new paragraph in section Results (from page 16 line 20 to page 17 line 17) and in few additional lines in the Discussion (page 22 lines 15-24). Unfortunately, including both pharmacodynamics and neural parameters in the same fitting procedure is not appropriate, due to the risk of a strong over-fitting. In our opinion, fitting neural parameters require additional tests, which should be the subject of future important research activity. The last aspect is declared at page 22 lines 9-14 in the Discussion.

---

## [Decision Letter · Decision Letter 1]

13 Feb 2020

Mathematical modeling and parameter estimation of levodopa motor response in patients with parkinson disease

PONE-D-19-25770R1

Dear Dr. Ursino,

We are pleased to inform you that your manuscript has been judged scientifically suitable for publication and will be formally accepted for publication once it complies with all outstanding technical requirements.

With kind regards,

Gennady Cymbalyuk, Ph.D.

Academic Editor

PLOS ONE

Additional Editor Comments (optional):

Reviewers' comments:

Reviewer's Responses to Questions

**Comments to the Author**

1. If the authors have adequately addressed your comments raised in a previous round of review and you feel that this manuscript is now acceptable for publication, you may indicate that here to bypass the “Comments to the Author” section, enter your conflict of interest statement in the “Confidential to Editor” section, and submit your "Accept" recommendation.

Reviewer #1: All comments have been addressed

Reviewer #2: All comments have been addressed

2. Is the manuscript technically sound, and do the data support the conclusions?

Reviewer #1: Yes

Reviewer #2: Yes

3. Has the statistical analysis been performed appropriately and rigorously? 

Reviewer #1: Yes

Reviewer #2: I Don't Know

4. Have the authors made all data underlying the findings in their manuscript fully available?

Reviewer #1: Yes

Reviewer #2: Yes

5. Is the manuscript presented in an intelligible fashion and written in standard English?

Reviewer #1: Yes

Reviewer #2: Yes

6. Review Comments to the Author

Reviewer #1: (No Response)

Reviewer #2: (No Response)

7. PLOS authors have the option to publish the peer review history of their article (what does this mean?). If published, this will include your full peer review and any attached files.

Reviewer #1: No

Reviewer #2: No

---

## [Editor Report · Acceptance letter]

19 Feb 2020

PONE-D-19-25770R1 

Mathematical modeling and parameter estimation of levodopa motor response in patients with parkinson disease 

Dear Dr. Ursino:

I am pleased to inform you that your manuscript has been deemed suitable for publication in PLOS ONE. Congratulations! Your manuscript is now with our production department. 

With kind regards,

on behalf of

Dr. Gennady Cymbalyuk 

Academic Editor

PLOS ONE